# Epidemiology, Clinical Characteristics, Risk Factors, and Outcomes of Candidemia in a Large Tertiary Teaching Hospital in Western China: A Retrospective 5-Year Study from 2016 to 2020

**DOI:** 10.3390/antibiotics11060788

**Published:** 2022-06-09

**Authors:** Jie Hou, Jin Deng, Ya Liu, Weili Zhang, Siying Wu, Quanfeng Liao, Ying Ma, Mei Kang

**Affiliations:** Division of Clinical Microbiology, Department of Laboratory Medicine, West China Hospital of Sichuan University, Chengdu 610041, China; microbe_houjie@hotmail.com (J.H.); dengjin@wchscu.cn (J.D.); liuya@wchscu.cn (Y.L.); zhangweili2046@sina.com (W.Z.); wusiying1015@163.com (S.W.); lqf114060@wchscu.cn (Q.L.)

**Keywords:** candidemia, *non-albicans Candida*, *Candida tropicalis*, *Candida albicans*, risk factors, azole resistance

## Abstract

The aim of this study was to investigate the current status of candidemia and evaluate the clinical characteristics, risk factors and outcomes among different species. We conducted a retrospective study by univariate and multivariate analysis between *Candida albicans* and *non-albicans Candida* (NAC) species in a Chinese national medical center from 2016 to 2020. Among the 259 episodes, *C. albicans* (38.6%) was the leading species, followed by *C. tropicalis* (24.3%), *C. parapsilosis* (20.5%), and *C. glabrata* (12.4%). Most *C. albicans* and *C. parapsilosis* were susceptible to nine tested antifungal agents, whereas *C. tropicalis* showed 30.2~65.9% resistance/non-wild-type to four azoles with great cross-resistance, indicating that fluconazole should not be used for empirical antifungal treatment. In multivariable models, the factor related to an increased risk of NAC was glucocorticoid exposure, whereas gastrointestinal hemorrhage and thoracoabdominal drainage catheters were associated with an increased risk in *C. albicans*. Subgroup analysis revealed leukemia and lymphoma, as well as glucocorticoid exposure, to be factors independently associated with *C. tropicalis* in comparison with *C. albicans* candidemia. No significant differences in 7-day mortality or 30-day mortality were observed between *C. albicans* and NAC. This study may provide useful information with respect to choosing empirical antifungal agents and exploring differences in molecular mechanisms.

## 1. Introduction

Candidemia is one of the leading causes of nosocomial bloodstream infections (BSI) and is a life-threatening invasive fungal infection associated with significant morbidity, mortality, high hospital costs, and successful clinical outcome that requires timely diagnosis and effective antifungal therapy [1,2]. Whereas the species distribution and susceptibility patterns of candidemia could vary considerably depending on geographic region and change over time, the global shift in favor of *non-albicans Candida* (NAC) species is of worldwide, as is the emerging and growing resistance to antifungal agents among these species [3]. Generally, *C. albicans* remains a leading causative agent of candidemia, but common NAC presents geographical variations, such as more familiar *C. tropicalis* in Asia and Latin America; more frequent *C. glabrata* in the USA and north/central Europe; and more prevalent *C. parapsilosis* in South America, southern Europe, and several parts of Asia. The top five *Candida* spp. account for approximately 90% of invasive candidiasis [1,3]. Given the varying patterns of susceptibility to azoles and echinocandins, changes in species distribution may drive the transformation of therapeutic and prophylactic strategies [4]. Intrinsic and acquired resistance to azoles in certain *Candida* spp. has posed considerable clinical challenges worldwide [5]. Echinocandins and fluconazole have been recommended as optional initial therapies for *Candida* infections according to clinical guidelines published by ESCMID [6] and IDSA [4].

Each *Candida* spp. Presents its own unique characteristics, including tissue tropism, invasive potential, virulence, and antifungal susceptibility [1,7]. To achieve improved benefits, early recognition and timely empirical antifungal treatment in populations at high risk of invasive fungal infections are essential, which require prediction of the drug resistance tendency of pathogens by taking into consideration local species distribution, antifungal susceptibility patterns, and species-related clinical features in individuals [1,4,6]. In this study, we reviewed the most recent microbial epidemiology of *Candida* antifungal resistance patterns by evaluating the existing circumstances of candidemia at West China Hospital of Sichuan University, a National Centre for Diagnosis and Treatment of Difficult and Critical Diseases, where approximately 4.85 million emergency and outpatient visits, as well as and 238 000 discharged patients were recorded in 2020. Furthermore, we compared *C. albicans* with NAC in terms of underlying conditions, possible risk factors, and clinical outcomes. We also performed a subgroup analysis of *C. albicans* vs. *C. tropicalis* to explore the key species-related variables in order to inform clinical decision making, providing interesting background on species-specific differences in terms of cellular and molecular factors.

## 2. Results

Between 2016 and 2020, a total of 667 positive blood cultures of *Candida* occurred in 316 patients at our hospital, as shown in Figure 1, and 57 patients were excluded. No significant variations in the proportion of *Candida* distribution were found, regardless of whether the mentioned patients were excluded. For the retrospective comparative study, we compared 100 patients with *C. albicans* candidemia with 159 patients with NAC candidemia and analyzed the differences in subgroups between *C. albicans* and *C. tropicalis.*

### 2.1. Species Distribution

Overall species distribution from 259 episodes of candidemia is depicted in Figure 2; eight different *Candida* spp. were identified. Among all *Candida*, the most frequent species was *C. albicans* (100/259, 38.6%), whereas NAC, including predominated *C. tropicalis* (63/259, 24.3%), *C. parapsilosis* (53/259, 20.5%), and *C. glabrata* (32/259, 12.4%) seemed to be the predominant causative agents for candidemia. The other uncommon NAC species accounted for less than 5% of all isolates, comprising *C. krusei* (5/259, 1.9%), *C. guilliermondii* (4/259, 1.5%), *C. lusitaniae* (1/259, 0.4%), and *C. haemulonii* (1/259, 0.4%). Figure 3 shows that the proportion and change trend of *C. albicans* and NAC did not change significantly over the study period from 2016 to 2020 (*p* = 0.744).

### 2.2. Antifungal Susceptibility Testing

The antifungal susceptibility testing (AFST) results of all isolates are presented in Table 1. Amphotericin B, flucytosine, and echinocandins demonstrated significant activity in vitro against most *Candida* spp. More than 96% of *Candida* were susceptible to three echinocandins, with the highest MIC_50_ of any species being ≤ 1 μg/mL and the highest MIC_90_ of any species being ≤2 μg/mL. Of the 259 *Candida* isolates, 175 (67.6%), 34 (13.1%), and 50 (19.3%) isolates were susceptible, susceptible dose-dependent (SDD), and resistant to fluconazole, respectively, and the main susceptible strains included *C. albicans* (92/100, 92.0%) and *C. parapsilosis* (45/53, 84.9%). The resistance rate of four azoles was the highest in *C. tropicalis*, with MIC_90_ values of 128, 2, 8, and 1 μg/mL to fluconazole, itraconazole, voriconazole, and posaconazole, respectively. Severe azole cross resistance was observed among *C. tropicalis*; most fluconazole-resistant *C. tropicalis* were non-susceptible (29/30, 96.7%) to voriconazole, and most voriconazole-resistant *C. tropicalis* were resistant (25/26, 96.2%) to fluconazole. Nonetheless, more than 91% of *C. albicans* were susceptible to four azoles, with an MIC_90_ ≤ 1 μg/mL.

### 2.3. Clinical Characteristics and Outcomes

Table 2 shows detailed demographic characteristics, underlying conditions and comorbidities, and clinical outcome variables. The median age was 53 (IQR: 43–66); elderly patients (age ≥ 65) accounted for 28.2% of the sample, and 31.7% of patients were female. Patients with *C. tropicalis* and other species of candidemia had a lower median age than those with *C. albicans* and *C. parapsilosis*. The majority of patients with candidemia were from intensive care units (ICUs) (45.6%), followed by medical wards (24.3%), surgical wards (13.1%), emergency departments (10.8%), and hematology wards (6.2%). The most common complications were gastrointestinal diseases (51.7%), lung diseases (51.4%), septic shock (32.8%), kidney diseases (28.2%), brain diseases (23.9%), liver diseases (22.8%), and solid tumors (19.3%). According to routine blood examinations (Table 3), almost all patients (93.1%) had different degrees of anemia, with insignificant differences among these species, whereas there were significant differences in platelets, white blood cells, neutrophil, and lymphocyte counts. Among patients with candidemia, the average total length of hospitalization was 32 days (IQR, 18–56) (Table 2). Patients with *C. tropicalis* or other candidemia species had a longer total hospitalization than those with *C. albicans* or *C. parapsilosis* candidemia, and a shorter ICU stay was found in *C. tropicalis or C. parapsilosis* candidemia than *C. albicans* or the other candidemia; however, these differences were not statistically significant. Moreover, no difference was found in 7-day mortality, 30-day mortality, or in-hospital mortality between the *Candida* spp. (Table 2), which is in line with the result of the survival curve (Figure 4).

### 2.4. C. albicans vs. Non-albicans Candida

Table 4 and Table 5 illustrate the details of potential risk factors associated with candidemia due to *C. albicans* and NAC, including comorbidities, common invasive procedures, and previous drug exposure, determined by univariable analysis and multivariate logistic regression, respectively. Compared with *C. albicans* candidemia, the prevalence of hematological disorders was markedly more frequent in patients with *C. tropicalis* candidemia, whereas the proportions of invasive procedures with significant differences, such as surgery, invasive mechanical ventilation, urinary catheters, and indwelling thoracoabdominal drainage catheters, were slightly lower in the NAC group. After the multivariate analysis presented in Table 5, several potential independent risk factors for candidemia with different *Candida* species were identified: glucocorticoids (OR 3.076, 95% CI 1.543–6.131, *p* = 0.001) were associated with NAC, whereas gastrointestinal hemorrhage (OR 0.397, 95% CI 0.209–0.755, *p* = 0.005) and thoracoabdominal drainage catheters (OR 0.507, 95% CI 0.289–0.891, *p* = 0.018) were closely related to *C. albicans*. Results of subgroup analysis of candidemia due to *C. tropicalis* or *C. albicans* to identify the factors associated with those two infections are presented in Table 6 and Table 7, respectively. Multivariate logistic regression analysis (Table 7) indicated leukemia and lymphoma (OR 10.08, 95%CI 1.127–90.133, *p* = 0.039), as well as glucocorticoids (OR 2.788, 95% CI 1.147–6.773, *p* = 0.024), as factors independently associated with *C. tropicalis* bloodstream infection, whereas thoracoabdominal drainage catheters (OR 0.277, 95% CI 0.131–0.588, *p* = 0.001) were separately connected with *C. albicans* candidemia according to previous logistic regression results between *C. albicans* vs. NAC (Table 5).

## 3. Discussion

In recent decades, an epidemiological trend shift from the dominant pathogen *C. albicans* to increasing incidence of NAC has been observed worldwide, although there is substantial geographic, center-to-center, and unit-to-unit variability in the relative prevalence of *Candida* spp. [1,3,8]. Likewise, we discovered *C. albicans* to be the most prevalent species, which is consistent with results from nationwide active laboratory-based surveillance in China [9]. The most common NAC identified in the current study was *C. tropicalis* (24.3%), which has been reported in several areas of similar latitudes [10] in contrast to reports from northern China [11], western Europe [12], and North America [13]. The distribution and frequency of *Candida* spp. Were influenced by not only geographic area but also the patient’s underlying conditions, the antifungal drugs patients had received, local hospital-related factors, and even the local climate [14,15]. The rates of these *Candida* spp. were similar in most diseases, but a significantly higher rate of *C. tropicalis* was observed in patients with hematologic malignancies who had undergone common cancer chemotherapy leading to neutropenia, which is in agreement with results of a prior study [16]. Additionally, antifungal exposure before the onset of candidemia might be partly accountable for the migration of *Candida* species to NAC [16], whereas a study in Thailand demonstrated that most patients with fluconazole-resistant *C. tropicalis* candidemia did not have a recent azole exposure and that *C. tropicalis* may represent exogenous isolates acquired from the environment [17].

In general, the tested antifungal agents, in addition to azoles, appeared to have a high susceptibility rate to common *Candida* according to CHIF-NET surveillance, which revealed the rapid emergence of azole-resistant *C. tropicalis* strains in China [18]. In our hospital, *C. albicans* and *C. parapsilosis* isolates from blood displayed significantly higher susceptibility or wild-type MIC to azoles than *C. tropicalis* and *C. glabrata* according to species-specific clinical breakpoints (CBPs) [19] and epidemiological cutoff values (ECVs) [20,21]. In contrast to the high susceptibility of *C. tropicalis* to amphotericin B, 5-flucytosine, and echinocandins, our study revealed relatively high azole resistance of *C. tropicalis* to four azoles, among which 47.6%, 41.3%, 30.2%, and 65.9% of isolates were resistant or had NWT MICs to fluconazole, voriconazole, itraconazole, and posaconazole, respectively. A higher MIC_90_ (128 μg/mL to fluconazole, 8 μg/mL to voriconazole) of azoles against *C. tropicalis* was found in our survey than in a previous investigation [18]. These important findings indicate that fluconazole and voriconazole should not be used as empirical antifungal drugs for treatment of *C. tropicalis*, which is in agreement with other studies [16,17]. Based on clinical practice guidelines [4,6] and our in vitro AFST data, echinocandins should be considered as the first choice for initial treatment of most episodes of candidemia and invasive candidiasis, except for central nervous system, eye, and urinary tract infections due to *Candida*. In addition, we observed an obvious cross resistance of *C. tropicalis* to fluconazole and voriconazole that may be related to different azole target Erg11p modifications or increased efflux pump activity [22]. On a larger scale, further study of the molecular mechanism of antifungal resistance, continuous antifungal resistance surveillance, development of non-cultured rapid diagnostic methods, and antifungal stewardship will be necessary to improve antifungal drug-resistant predicaments [23].

The most common individual predisposing factors for invasive candidiasis include those intrinsic to the host or the disease state, such as *Candida* colonization, old age, diabetes mellitus, gastrointestinal perforation, pancreatitis, sepsis, hematologic malignancy, neutropenia, transplantation, and severe immunodeficiency, as well as factors resulting from iatrogenic interventions, such as long-term and/or repeated use of broad-spectrum antibiotics, recent major surgery (particularly abdominal surgery), dialysis, parenteral nutrition (PN), use of corticosteroids, use of immunosuppressants, presence of indwelling central venous catheters, and long-term ICU stays [1,4,24]. In the current study, most patients had been previously exposed to antibiotics (>95%) and PN (>70%) without significant interspecies differences, which indicates that these may be common contributing factors for candidemia. Broad-spectrum antibiotics could confer *Candida* spp. a selective advantage over bacteria, causing *Candida* spp. overgrowth and increased gut colonization [1]. Parenteral lipid emulsion could increase *Candida* biofilm formation, which may explain the increased risk of candidemia in patients receiving parenteral nutrition via medical catheters [25]. Each *Candida* spp. presents its own unique characteristics, including tissue tropism, invasive potential, virulence, biofilm formation ability, and antifungal susceptibility [1,4,25].

Several studies have compared the characteristics of *C. albicans* with those of NAC candidemia, revealing differences in risk factors and outcomes [11,16,17,26,27,28,29]. Multivariate analysis also confirmed that glucocorticoids exposure is associated with an increased risk of NAC candidemia, which is consistent with results of other studies [29]. We also found an association of gastrointestinal hemorrhage and indwelling thoracoabdominal drainage catheters with a higher risk of *C. albicans* candidemia, which is in agreement with results reported by Gong et al. [30], which may be attributed to *C. albicans* being the most frequent *Candida* species in the human gut mycobiome [31]. Some studies led to different, conflicting conclusions; for example, one risk factor, PN, was linked to a decreased risk of NAC candidemia by Chow et al. [28], but Zhang et al. suggested that PN was associated with an increased risk of NAC [11], and there was no significant difference among species in this cohort. The possible reason for such paradoxical conclusions may be associated with variability of *Candida* species distribution, as well as patients’ baseline comorbidities in the local epidemiological setting.

Risk factors associated with candidemia caused by *C. tropicalis* or *C. albicans* were also compared in a subgroup analysis. The results of that analysis revealed that leukemia and lymphoma, as well as glucocorticoid exposure, are independent risk factors for *C. tropicalis* candidemia. A previous study reported the independent risk factors for *C. tropicalis* bloodstream infections as neutropenia, chronic liver disease, and male sex [17]. Our study confirmed that *C. tropicalis* seems to be more frequent in patients with hematological diseases (neutropenia, leukemia, and lymphoma), which is in agreement with previous studies suggesting that *C. tropicalis* is the most common *Candida* species (75.4%), rather than *C. albicans* (12.3%), in hematological malignancy [32], which may be linked with cytotoxic chemotherapy-induced immunosuppression.

Several investigations have recorded discrepant results of clinical outcomes between *C. albicans* and NAC candidemia. Although some investigations have reported that mortality was higher in patients with NAC than in those with *C. albicans* candidemia [11,29], insignificant differences in 7-day mortality, 30-day mortality, and in-hospital mortality were discovered among different *Candida* species groups in the present study (Table 2 and Figure 4). On the one hand, early mortality is linked with prompt therapeutic measures, including appropriate antifungal agents and early removal of intravascular catheters, as recommended in guidelines [4,33]; on the other hand, late mortality is associated with host factors (e.g., patient comorbid status and signs of organ dysfunction) [33]. The few significant differences in mortality observed in our study could be partly due to the choice preference for echinocandin use in our setting, as well as the similar severity of underlying conditions among patients (Table 2).

Several limitations need to be noted in our study. First, the results of a retrospective study can be exploratory and should be interpreted with caution. Second, two AFST methods were applied at different times, so we lacked data on posaconazole and echinocandins for few months. Furthermore, the study was performed at a center lacking pediatrics, obstetrics, and gynecology disciplines; thus, the results may not be applicable to other settings.

## 4. Materials and Methods

### 4.1. Setting, Study Design, and Data Collection

This research was conducted as a retrospective epidemiological investigation and an attendant comparative study from 2016 to 2020 at the West China Hospital of Sichuan University, a tertiary grade A academic teaching hospital with 4300 beds in Chengdu. All positive *Candida* blood cultures were identified from the microbiological laboratory information system. The electronic medical records of all patients were retrospectively reviewed, and the following information was collected from the hospital information system: age, sex, diagnosis at admission and discharge, comorbidities, prior use of drugs, invasive procedures, laboratory products, and clinical outcomes. The Charlson Comorbidity Index (CCI) and age-adjusted Charlson Comorbidity Index (aCCI) were calculated for each case to assess the severity of illness.

### 4.2. Definitions

Candidemia was defined by at least one positive blood culture for *Candida* spp. in patients with compatible clinical signs and symptoms of infection. Only the first case of positive *Candida* blood cultures was included, and patients < 16 years of age, mixed candidemia or outpatient, and emergency patients with incomplete clinical information were excluded. The onset of candidemia was defined as the date when the first positive blood culture specimen was collected. Prior invasive procedures and drug exposure are defined as occurrence of the relevant event within 30 days before the onset of candidemia.

### 4.3. Microbiological Analysis

Blood specimens were processed using a Bact/Alert 3D automated blood culture system (bioMérieux, Marcyl’Etoile, France). Colonies of *Candida* isolates were identified with either matrix-assisted laser desorption ionization time-of-flight mass spectrometry (Bruker Daltoniks, Bremen, Germany) or internal transcribed spacer sequencing. All analyzed isolates had AFST performed using ATB FUNGUS three strips (bioMérieux, Marcyl’Etoile, France) from January 2016 to May 2017 and were performed using Sensititre YeastOne^TM^ commercialized products (Trek Diagnostic Systems Ltd., East Grimstead, UK) consisting of nine antifungal agents after April 2017. *C. krusei* ATCC 6258 and *C. parapsilosis* ATCC 22019 were routinely used as quality controls. AFST was interpreted using species-specific CBPs defined by the Clinical Laboratory Standards Institute in the M60 [19] or ECVs in the M59 [20] and previous studies [21] to distinguish wild-type (WT) or non-wild-type (NWT) strains where CBPs were not available. The minimum inhibitory concentration (MIC) distribution of each *Candida* spp. was summarized to obtain MIC_50_ and MIC_90_.

### 4.4. Statistical Analysis

Quantitative variables reported as the median and interquartile range (IQR) were analyzed with a Mann–Whitney test or Kruskal–Wallis test where applicable. Categorical variables presented as absolute numbers and relative percentages were compared between groups with a chi-square test or Fisher’s exact test as appropriate. Non-colinear covariates with a *p* value ≤ 0.05 in the univariate analysis after deliberation based on practical clinical significance were applied in stepwise logistic regression multivariate model to identify independent factors, and the results were presented as odds ratios (OR) with their 95% confidence intervals (95% CI) and *p* values. The 30-day survival curves of candidemia were delineated by Kaplan–Meier survival analysis, and the difference was evaluated by the log-rank test. Linear-by-linear association analyses were performed to evaluate a changing trend in species distribution patterns over the five-year study period. All significance tests were two-tailed. P values of 0.05 or less were considered boundaries for statistical significance. All data were analyzed by IBM SPSS 26.0 (IBM, Armonk, NY, USA).

## 5. Conclusions

In conclusion, we found that *C. tropicalis* was the most common NAC species, with a particularly alarming resistance to azoles among candidemia in western China over the past 5 years, so empirical treatment would not recommend using azoles. NAC candidemia was more frequent than *C. albicans* in patients who had been exposed to glucocorticoids in contrast to patients with gastrointestinal hemorrhage and indwelling thoracoabdominal drainage catheters. Continued and careful monitoring of the clinical and mycological characteristics of candidemia is required, as well as further evaluation in clinical practice and further in-depth research.

## Figures and Tables

**Figure 1 antibiotics-11-00788-f001:**
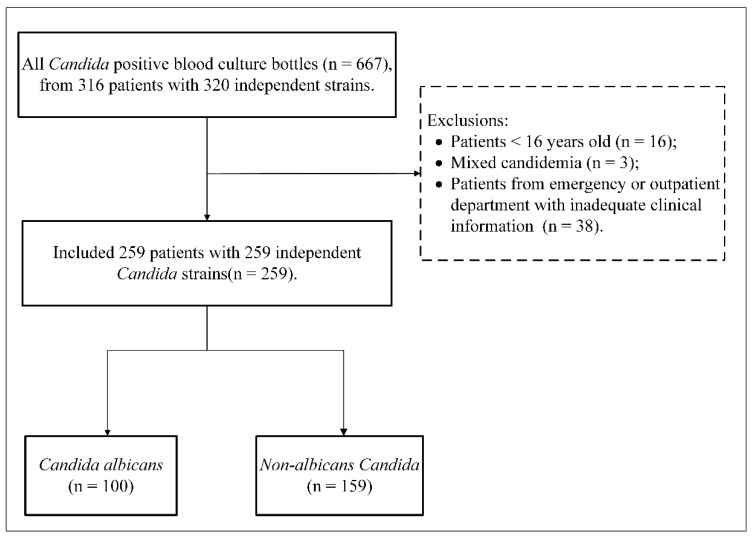
Flow chart of the comparative study from 2016 to 2020.

**Figure 2 antibiotics-11-00788-f002:**
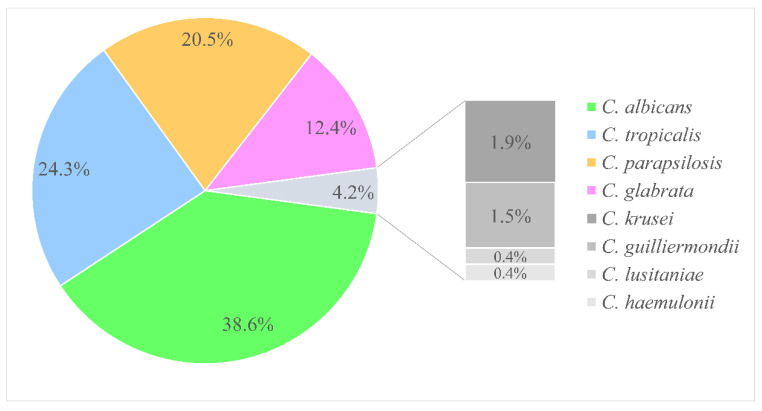
Distribution of *Candida* species in candidemia from 2016 to 2020 (*n* = 259).

**Figure 3 antibiotics-11-00788-f003:**
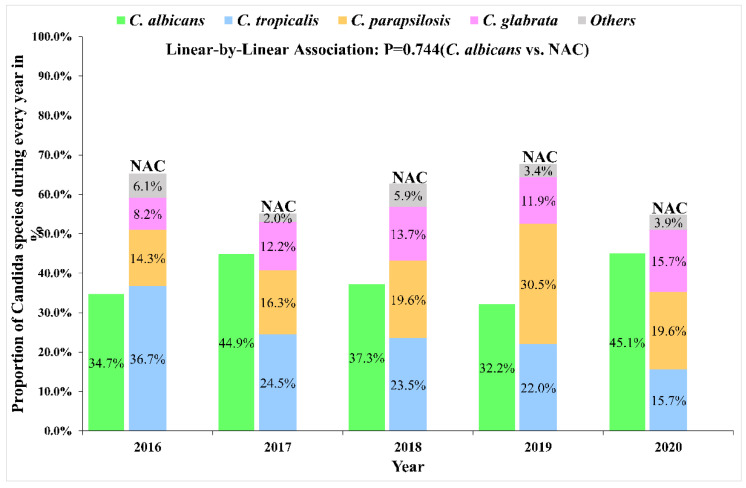
Change trend of *C. albicans* vs. *non-albicans Candida* over the five-year study period (*n* = 259).

**Figure 4 antibiotics-11-00788-f004:**
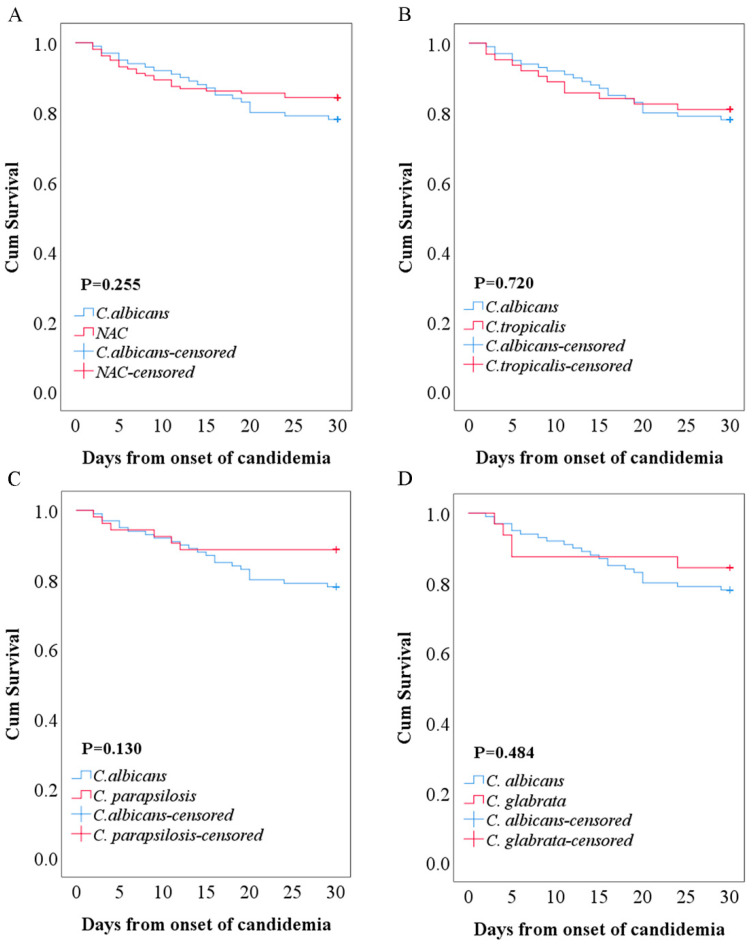
Kaplan–Meier survival curve of patients with *C. albicans* and *non-albicans* candidemia. Survival curve for 100 *C. albicans* vs. 159 *non-albicans Candida* candidemia (**A**), 100 *C. albicans* vs. 63 *C. tropicalis* candidemia (**B**), 100 *C. albicans* vs. 53 *C. parapsilosis* candidemia (**C**), 100 *C. albicans* vs. 32 *C. glabrata* candidemia (**D**).

**Table 1 antibiotics-11-00788-t001:** Antifungal susceptibility of *Candida* species in bloodstream infections from 2016 to 2020.

*Candida* spp.	Antifungal ^a^	MIC (μg/mL)			Antifungal Susceptibility		
		Ranges	MIC_50_	MIC_90_	S/WT %	SDD/I %	R/NWT%
*C. albicans*	Amphotericin B	0.12–1	0.5	0.5	100(100%)	-	0
	5-Flucytosine	0.008–4	0.015	4	100(100%)	-	0
	Fluconazole	0.12–256	0.5	1	92(92.0%)	2(2.0%)	6(6.0%)
	Voriconazole	0.008–8	0.008	0.125	92(92.0%)	4(4.0%)	4(4.0%)
	Itraconazole	0.008–16	0.06	0.125	91(91.0%)	-	9(9.0%)
	Posaconazole	≤0.008–8	0.015	0.03	72(93.5%)	-	5(6.5%)
	Anidulafungin	0.015–0.5	0.03	0.12	77(100%)	0	0
	Caspofungin	0.008–0.25	0.06	0.12	77(100%)	0	0
	Micafungin	≤0.008–0.5	0.008	0.015	77(100%)	0	0
*C. tropicalis*	Amphotericin B	0.25–1	0.5	1	63(100%)	-	0
	5-Flucytosine	0.008–4	0.015	4	63(100%)	-	0
	Fluconazole	0.25–256	1	128	32(50.8%)	1(1.6%)	30(47.6%)
	Voriconazole	0.008–8	0.25	8	31(49.2%)	6(9.5%)	26(41.3%)
	Itraconazole	0.06–16	0.5	2	44(69.8%)	-	19(30.2%)
	Posaconazole	0.015–8	0.25	1	14(34.1%)	-	27(65.9%)
	Anidulafungin	0.015–0.5	0.12	0.25	39(95.1%)	2(4.9%)	0
	Caspofungin	0.015–0.25	0.06	0.25	41(100%)	0	0
	Micafungin	0.015–0.5	0.03	0.06	40(97.6%)	1(2.4%)	0
*C. parapsilosis*	Amphotericin B	0.12–1	0.25	0.5	53(100%)	-	0
	5-Flucytosine	0.008–4	0.015	4	53(100%)	-	0
	Fluconazole	0.12–128	0.5	4	45(84.9%)	5(9.4%)	3(5.7%)
	Voriconazole	0.008–2	0.015	0.06	49(92.5%)	0	4(7.5%)
	Itraconazole	0.015–0.5	0.03	0.125	53(100%)	-	0
	Posaconazole	0.008–0.5	0.015	0.06	45(100%)	-	0
	Anidulafungin	0.12–2	1	2	45(100%)	0	0
	Caspofungin	0.008–2	0.5	0.5	45(100%)	0	0
	Micafungin	0.03–2	1	1	45(100%)	0	0
*C. glabrata*	Amphotericin B	0.12–2	0.5	2	32(100%)	-	0
	5-Flucytosine	0.008–16	0.008	4	31(96.9%)	-	1(3.1%)
	Fluconazole	0.25–256	4	64	-	26(81.2%)	6(18.8%)
	Voriconazole	0.008–4	0.125	1	18(56.3%)	-	14(43.8%)
	Itraconazole	0.015–16	0.500	1	29(90.6%)	-	3(9.4%)
	Posaconazole	0.015–8	0.5	8	17(65.4%)	-	9(34.6%)
	Anidulafungin	0.125–2	0.03	0.06	24(92.3%)	0	2(7.7%)
	Caspofungin	0.008–0.5	0.03	0.25	21(80.8%)	3(11.5%)	2(7.7%)
	Micafungin	0.008–1	0.015	0.03	24(92.3%)	0	2(7.7%)
Others ^b^	Amphotericin B	0.12–1	0.25	0.5	11(100%)	-	0
	5-Flucytosine	0.008–16	1	4	11(100%)	-	0
	Fluconazole	0.12–256	4	64	6(54.5%)	0	5(45.5%)
	Voriconazole	0.008–0.5	0.12	0.25	11(100%)	0	0
	Itraconazole	0.03–1	0.25	0.5	11(100%)	-	0
	Posaconazole	0.015–0.5	0.12	0.25	7(100%)	-	0
	Anidulafungin	0.015–8	0.25	2	5(71.4%)	0	2(28.6%)
	Caspofungin	0.015–8	0.5	2	4(57.1%)	1(14.3%)	2(28.6%)
	Micafungin	0.015–4	0.12	2	4(57.1%)	2(28.6%)	1(14.3%)

^a^: ATB FUNGUS 3 strips (bioMérieux, Marcyl’Etoile, France) lacking posaconazole, caspofungin, micafungin and anidulafungin were performed from January 2016 to May 2017; Sensititre YeastOne^TM^ antifungal panels (Trek Diagnostic Systems Ltd., East Grimstead, UK) consisting of nine antifungal agents were performed after April 2017. Therefore, data on posaconazole and echinocandins were lacking for a few months. ^b^: Other *Candida* spp. include five *Candida krusei* (five isolates of *C. krusei* were inherently resistant to fluconazole, one *C. krusei* strain was intermediate to caspofungin, and another was intermediate to micafungin), four *Candida guilliermondii* (one *C. guilliermondii* strain was resistant to anidulafungin and caspofungin and intermediate to micafungin), one *Candida lusitaniae* (one wild-type isolate for nine tested agents), and one *Candida haemulonii* (this strain was considered non-wild type for echinocandins, as its MIC of each echinocandin is 2 μg/mL; however, it had low MIC values as a WT isolate for other antifungals with the following contents: amphotericin B: 0.250 μg/mL, 5-flucytosine: 0.008 μg/mL, fluconazole: 0.5 μg/mL, itraconazole: 0.03 μg/mL, voriconazole: 0.008 μg/mL, posaconazole: 0.03 μg/mL).Abbreviations: MIC: minimum inhibitory concentration; MIC_50_: 50% minimum inhibitory concentration; MIC_90_: 90% minimum inhibitory concentration; S: susceptible; SDD: susceptible dose-dependent; I: intermediate; R: resistance; WT: wild type; NWT: non-wild type.

**Table 2 antibiotics-11-00788-t002:** Baseline characteristics of patients with *C. albicans* and NAC candidemia.

Characteristic	Total (*n* = 259)	*C. albicans* (*n* = 100)	*C. tropicalis* (*n* = 63)	*C. parapsilosis* (*n* = 53)	Others ^a^ (*n* = 43)	*p* Value
Age	53(43–66)	56(46–67)	50(33–63)	56(44–68)	47(30–63)	0.008
Age (≥65 years)	73(28.2%)	34(34.0%)	14(22.2%)	16(30.2%)	9(20.9%)	0.261
Female	82(31.7%)	34(34.0%)	18(28.6%)	16(30.2%)	14(32.6%)	0.896
Length of hospital stay	32(18–56)	29(19–49.5)	39(21–62)	27(15–49)	32(17.5–67.5)	0.378
Length of ICU stay	27(14–46)	29(20–42)	21(13–42)	22(10–49)	32(11–55)	0.329
Previous ICU stay	139(53.7%)	55(55.0%)	31(49.2%)	30(56.6%)	23(53.5%)	0.859
**Medical service**						
ICUs	118(45.6%)	51(51.0%)	24(38.1%)	22(41.5%)	21(48.8%)	-
Medical wards	63(24.3%)	26(26.0%)	14(22.2%)	18(34.0%)	5(11.6%)	-
Surgical wards	34(13.1%)	14(14.0%)	5(7.9%)	7(13.2%)	8(18.6%)	-
Emergency departments	28(10.8%)	8(8.0%)	8(12.7%)	5(9.4%)	7(16.3%)	-
Hematology wards	16(6.2%)	1(1.0%)	12(19.0%)	1(1.9%)	2(4.7%)	-
**Underlying conditions**						
CCI	2(1–3)	2(1–3)	2(1–4)	2(1–4)	2(0–4)	0.480
aCCI	3(2–5)	3(1–5)	3(2–5)	4(2–6)	3(1–5)	0.334
Gastrointestinal diseases	134(51.7%)	52(52.0%)	26(41.3%)	29(54.7%)	27(62.8%)	0.167
Lung diseases	133(51.4%)	52(52.0%)	27(42.9%)	31(58.5%)	23(53.5%)	0.392
Septic shock	85(32.8%)	40(40.0%)	18(28.6%)	13(24.5%)	14(32.6%)	0.212
Kidney diseases	73(28.2%)	28(28.0%)	14(22.2%)	18(34.0%)	13(30.2%)	0.558
Brain diseases	62(23.9%)	31(31.0%)	11(17.5%)	11(20.8%)	9(20.9%)	0.195
Liver diseases	59(22.8%)	22(22.0%)	15(23.8%)	16(30.2%)	6(14.0%)	0.304
Solid tumors	50(19.3%)	18(18.0%)	12(19.0%)	9(17.0%)	11(25.6%)	0.710
Diabetes mellitus	47(18.1%)	21(21.0%)	10(15.9%)	6(11.3%)	10(23.3%)	0.364
Heart diseases	36(13.9%)	16(16.0%)	5(7.9%)	11(20.8%)	4(9.3%)	0.166
Neutropenia	24(9.3%)	4(4.0%)	15(23.8%)	2(3.8%)	3(7.0%)	0.000
Leukemia and Lymphoma	17(6.6%)	1(1.0%)	12(19.0%)	2(3.8%)	2(4.7%)	0.000
Transplantation	10(3.9%)	2(2.0%)	6(9.5%)	1(1.9%)	1(2.3%)	0.102
**Outcomes**						
7-day mortality	20(7.7%)	6(6.0%)	5(7.9%)	3(5.7%)	6(14.0%)	0.392
30-day mortality	47(18.1%)	22(22.0%)	12(19.0%)	6(11.3%)	7(16.3%)	0.424
In-hospital mortality ^b^	56(21.6%)	23(23.0%)	18(28.6%)	6(11.3%)	9(20.9%)	0.128

^a^: Others included 32 *Candida glabrata*, 5 *Candida krusei*, 4 *Candida guilliermondii*, 1 *Candida lusitaniae* and 1 *Candida haemulonii*. ^b^: There is an overlap in 7-day mortality, 30-day mortality, and in-hospital mortality. Abbreviations: NAC, *non-albicans Candida*; ICU, intensive care unit; CCI, Charlson Comorbidity Index; aCCI, age-adjusted Charlson Comorbidity Index.

**Table 3 antibiotics-11-00788-t003:** Clinical laboratory data of patients with *C. albicans* and NAC candidemia.

Characteristic	Total (*n* = 259)	*C. albicans* (*n* = 100)	*C. tropicalis* (*n* = 63)	*C. parapsilosis* (*n* = 53)	Others ^a^ (*n* = 43)	*p* Value
RBC (×10^12^/L)	2.96(2.50–3.50)	2.91(2.50–3.60)	2.84(2.37–3.24)	3.15(2.59–3.51)	3.00(2.50–3.41)	0.331
HGB (g/L)	86(75–100)	86(75–105)	83(71–99)	89(80–103)	86(75–96)	0.298
PLT (×10^9^/L)	127(67–230)	153(73–260)	87(19–159)	139(94–220)	141(69–230)	0.003
WBC (×10^9^/L)	8.86(5.01–13.00)	10.86(7.67–14.21)	8.76(2.19–13.00)	5.89(3.91–8.95)	8.98(5.12–13.40)	0.000
Neutrophils (×10^9^/L)	7.47(3.99–11.30)	9.34(6.54–12.52)	6.70(0.64–11.66)	4.43(3.01–7.37)	7.34(4.49–11.93)	0.000
Lymphocyte (×10^9^/L)	0.60(0.38–1.03)	0.62(0.43–1.02)	0.59(0.23–0.88)	0.76(0.47–1.23)	0.48(0.33–0.86)	0.026
Monocyte (×10^9^/L)	0.38(0.15–0.57)	0.42(0.20–0.61)	0.32(0.05–0.56)	0.38(0.20–0.53)	0.42(0.10–0.66)	0.174
Anemia	241(93.1%)	92(92.0%)	58(92.1%)	49(92.5%)	42(97.7%)	0.661
Leukopenia	43(16.6%)	5(5.0%)	19(30.2%)	12(22.6%)	7(16.3%)	0.000
Leukocytosis	115(44.4%)	61(61.0%)	25(39.7%)	11(20.8%)	18(41.9%)	0.000
Thrombocytopenia	98(37.8%)	34(34.0%)	34(54.0%)	14(26.4%)	16(37.2%)	0.014
Thrombocytosis	26(10.0%)	16(16.0%)	3(4.8%)	3(5.7%)	4(9.3%)	0.084
Hepatitis B virus	30(11.6%)	11(11.0%)	10(15.9%)	4(7.5%)	5(11.6%)	0.573
Tuberculosis	12(4.6%)	3(3.0%)	2(3.2%)	5(9.4%)	2(4.7%)	0.341
Fluconazole NS	84(32.4%)	8(8.0%)	31(49.2%)	8(15.1%)	37(86.0%)	0.000
Voriconazole NS	58(22.4%)	8(8.0%)	32(50.8%)	4(7.5%)	14(32.6%)	0.000
Azole cross resistance	53(20.5%)	6(6.0%)	30(47.6%)	3(5.7%)	14(32.6%)	0.000

^a^: Others included 32 *Candida glabrata*, 5 *Candida krusei* (*C. krusei* were inherently resistant to fluconazole), 4 *Candida guilliermondii*, 1 *Candida lusitaniae*, and 1 *Candida haemulonii*. Abbreviations: NAC, *non-albicans Candida*; RBC, red blood cells; HGB, hemoglobin; PLT, platelets; WBC, white blood cells; NS, non-susceptible.

**Table 4 antibiotics-11-00788-t004:** Univariate analysis for risk factors regarding patients with *C. albicans* vs. NAC candidemia.

Characteristic	Total (*n* = 259)	*C. albicans* (*n* = 100)	NAC (*n* = 159)	*p* Value
Age (≥65 years)	73(28.2%)	34(34.0%)	39(24.5%)	0.099
Gastrointestinal hemorrhage	54(20.8%)	30(30.0%)	24(15.1%)	0.004
Intra-abdominal infections	54(20.8%)	27(27.0%)	27(17.0%)	0.053
Pancreatitis	50(19.3%)	22(22.0%)	28(17.6%)	0.384
Peritonitis	31(12.0%)	17(17.0%)	14(8.8%)	0.048
Gastrointestinal perforation	25(9.7%)	12(12.0%)	13(8.2%)	0.310
Septic shock	85(32.8%)	40(40.0%)	45(28.3%)	0.051
Solid tumors	50(19.3%)	18(18.0%)	32(20.1%)	0.673
Diabetes mellitus	47(18.1%)	21(21.0%)	26(16.4%)	0.345
Urinary tract infections	37(14.3%)	12(12.0%)	25(15.7%)	0.404
Neutropenia	24(9.3%)	4(4.0%)	20(12.6%)	0.020
Leukemia and lymphoma	17(6.6%)	1(1.0%)	16(10.1%)	0.004
Transplantation	10(3.9%)	2(2.0%)	8(5.0%)	0.325
Parenteral nutrition	183(70.7%)	72(72.0%)	111(69.8%)	0.706
Central venous catheters	152(58.7%)	68(68.0%)	84(52.8%)	0.016
Urinary catheters	140(54.1%)	65(65.0%)	75(47.2%)	0.005
Previous ICU Stay	139(53.7%)	55(55.0%)	84(52.8%)	0.733
Invasive mechanical ventilation	138(53.3%)	64(64.0%)	74(46.5%)	0.006
Thoracoabdominal drainage Catheters	119(45.9%)	59(59.0%)	60(37.7%)	0.001
Surgery	114(44.0%)	54(54.0%)	60(37.7%)	0.010
Abdominal surgery	69(26.6%)	32(32.0%)	37(23.3%)	0.122
Hemodialysis	47(18.1%)	15(15.0%)	32(20.1%)	0.297
Antibiotics	247(95.4%)	94(94.0%)	153(96.2%)	0.545
Carbapenems	125(48.3%)	48(48.0%)	77(48.4%)	0.947
Piperacillin	80(30.9%)	26(26.0%)	54(34.0%)	0.177
Cephalosporins (third and fourth)	77(29.7%)	27(27.0%)	50(31.4%)	0.446
Cefoperazone-sulbactam	58(22.4%)	21(21.0%)	37(23.3%)	0.670
Quinolones	51(19.7%)	15(15.0%)	36(22.6%)	0.132
Vancomycin	46(17.8%)	20(20.0%)	26(16.4%)	0.455
Tigecycline	32(12.4%)	12(12.0%)	20(12.6%)	0.890
Antifungal drugs	38(14.7%)	11(11.0%)	27(17.0%)	0.185
Azoles	29(11.2%)	9(9.0%)	20(12.6%)	0.374
Echinocandins	11(4.2%)	3(3.0%)	8(5.0%)	0.538
Glucocorticoids	65(25.1%)	4(4.0%)	25(15.7%)	0.001
Cancer chemotherapy	52(20.1%)	12(12.0%)	40(25.2%)	0.010
Immunosuppressants	15(5.8%)	14(14.0%)	51(32.1%)	0.127

Abbreviations: NAC, *non-albicans Candida*; ICU, intensive care units; Cephalosporins (third and fourth), third-generation and fourth-generation cephalosporins.

**Table 5 antibiotics-11-00788-t005:** Multivariate analysis of possible risk factors regarding *C. albicans* vs. NAC candidemia.

Variable	Unadjusted OR (95% CI)	*p* Value	Adjusted OR (95% CI)	*p* Value
Gastrointestinal hemorrhage	0.415(0.226–0.763)	0.005	0.397(0.209–0.755)	0.005
Surgery	0.516(0.311–0.858)	0.011	0.609(0.346–1.073)	0.086
Thoracoabdominal drainage catheters	0.421(0.253–0.702)	0.001	0.507(0.289–0.891)	0.018
Glucocorticoids	2.901(1.506–5.588)	0.001	3.076(1.543–6.131)	0.001

Note: Variables were entered in the first step using a backward stepwise method, including gastrointestinal hemorrhage, peritonitis, leukemia and lymphoma, neutropenia, surgery, invasive mechanical ventilation, central venous catheters, urinary catheters, thoracoabdominal drainage catheters, glucocorticoids, and cancer chemotherapy. Abbreviations: NAC, *non-albicans Candida*; OR, odds ratio; CI, confidence interval.

**Table 6 antibiotics-11-00788-t006:** Univariate analysis of factors with respect to patients with *C. albicans* vs. *C. tropicalis* candidemia.

Characteristic	Total (*n* = 163)	*C. albicans* (*n* = 100)	*C. tropicalis* (*n* = 63)	*p* Value
Age (≥65 years)	48(29.4%)	34(34.0%)	14(22.2%)	0.108
Gastrointestinal hemorrhage	43(26.4%)	30(30.0%)	13(20.6%)	0.186
Intra-abdominal infections	36(22.1%)	27(27.0%)	9(14.3%)	0.057
Pancreatitis	35(21.5%)	22(22.0%)	13(20.6%)	0.836
Peritonitis	19(11.7%)	17(17.0%)	2(3.2%)	0.007
Gastrointestinal perforation	13(8.0%)	12(12.0%)	1(1.6%)	0.017
Septic shock	58(35.6%)	40(40.0%)	18(28.6%)	0.138
Diabetes mellitus	31(19.0%)	21(21.0%)	10(15.9%)	0.417
Solid tumors	30(18.4%)	18(18.0%)	12(19.0%)	0.867
Urinary tract infections	22(13.5%)	12(12.0%)	10(15.9%)	0.481
Neutropenia	19(11.7%)	4(4.0%)	15(23.8%)	0.000
Leukemia and lymphoma	13(8.0%)	1(1.0%)	12(19.0%)	0.000
Transplantation	8(4.9%)	2(2.0%)	6(9.5%)	0.056
Parenteral nutrition	112(68.7%)	72(72.0%)	40(63.5%)	0.254
Central venous catheters	98(60.1%)	68(68.0%)	30(47.6%)	0.010
Urinary catheters	95(58.3%)	65(65.0%)	30(47.6%)	0.028
Invasive mechanical ventilation	91(55.8%)	64(64.0%)	27(42.9%)	0.008
Previous ICU stay	86(52.8%)	55(55.0%)	31(49.2%)	0.471
Thoracoabdominal drainage Catheters	74(45.4%)	59(59.0%)	15(23.8%)	0.000
Surgery	73(44.8%)	54(54.0%)	19(30.2%)	0.003
Abdominal surgery	42(25.8%)	32(32.0%)	10(15.9%)	0.022
Hemodialysis	26(16.0%)	15(15.0%)	11(17.5%)	0.676
Antibiotics	154(94.5%)	94(94.0%)	60(95.2%)	1.000
Carbapenems	84(51.5%)	48(48.0%)	36(57.1%)	0.255
Piperacillin	46(28.2%)	26(26.0%)	20(31.7%)	0.427
Cephalosporins (third and fourth)	47(28.8%)	27(27.0%)	20(31.7%)	0.515
Cefoperazone-sulbactam	35(21.5%)	21(21.0%)	14(22.2%)	0.853
Quinolones	31(19.0%)	15(15.0%)	16(25.4%)	0.100
Vancomycin	30(18.4%)	20(20.0%)	10(15.9%)	0.508
Tigecycline	20(12.3%)	12(12.0%)	8(12.7%)	0.895
Antifungal drugs	24(14.7%)	11(11.0%)	13(20.6%)	0.091
Azoles	17(10.4%)	9(9.0%)	8(12.7%)	0.452
Echinocandins	6(3.7%)	3(3.0%)	3(4.8%)	0.677
Glucocorticoids	39(23.9%)	14(14.0%)	25(39.7%)	0.000
Cancer chemotherapy	32(19.6%)	12(12.0%)	20(31.7%)	0.002
Immunosuppressants	11(6.7%)	3(3.0%)	8(12.7%)	0.023

Abbreviations: ICU, Intensive care units; Cephalosporins (third and fourth), third-generation and fourth-generation cephalosporins.

**Table 7 antibiotics-11-00788-t007:** Multivariate analysis of factors with respect to patients with *C. albicans* vs. *C. tropicalis* candidemia.

Variable	Unadjusted OR (95% CI)	*p* Value	Adjusted OR (95% CI)	*p* Value
Gastrointestinal perforation	0.118(0.015–0.933)	0.043	0.193(0.022–1.689)	0.137
Thoracoabdominal drainage catheters	0.217(0.107–0.439)	0.000	0.277(0.131–0.588)	0.001
Leukemia and lymphoma	23.294(2.946–184.192)	0.003	10.08(1.127–90.133)	0.039
Glucocorticoids	4.041(1.895–8.620)	0.000	2.788(1.147–6.773)	0.024

Note: Variables were entered in the first step using a backward stepwise method, including gastrointestinal perforation, peritonitis, leukemia and lymphoma, neutropenia, surgery, abdominal surgery, invasive mechanical ventilation, central venous catheters, urinary catheters, thoracoabdominal drainage catheters, immunosuppressants, glucocorticoids, and cancer chemotherapy. Abbreviations: OR, odds ratio; CI, confidence interval.

## Data Availability

The data and material information used and analyzed in the current study are available from the corresponding author upon reasonable request.

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
