# Peer review of "Epidemiology, Clinical Characteristics, Risk Factors, and Outcomes of Candidemia in a Large Tertiary Teaching Hospital in Western China: A Retrospective 5-Year Study from 2016 to 2020"

_antibiotics, 2022, doi:10.3390/antibiotics11060788_

Round 1

Reviewer 1 Report

In the manuscript by Hou et. al., the authors conducted a retrospective study to assess the risk factors and outcomes of Candida infections in a hospital. The study was conducted on the data obtained from 259 patients. It is well planned, and the outcomes are useful for developing successful strategies for controlling Candida infections. However, there are a few comments which need to be addressed.

  1. The scientific names should be italicized throughout the manuscript.
  2. All figures are too small and not legible to read. High-resolution images with greater size should be provided.
  3. Line 52: Should read as ‘we attempted to review…’
  4. Line 87: should read as ‘demonstrated significant activity…’. Remove the word ‘brilliant’
  5. Table 2: How was the ‘in-hospital mortality’ different from 7-day and 30-day mortality. Is there an overlap in these numbers? It is surprising that nearly 51% of the patients infected with C. albicans died. This number is too high compared to reported statistics on the Candidemia death rate. The authors should further explain this significant number of deaths. I am assuming that the death rate is high because of underlying causes which were aggravated by Candidemia. Please explain.
  6. Lines 176-179: delete these lines.

Author Response

Thanks very much for taking your time to review the previous version of the manuscript (Manuscript ID: antibiotics-1712183 and Title: Epidemiology, clinical characteristics, risk factors and outcomes of candidemia in a large tertiary teaching hospital in western China: a retrospective 5-year study from 2016 to 2020). We really appreciate all your comments and suggestions! These comments are all valuable and very helpful for revising and improving our paper, as well as the important guiding significance to our research. We have carefully considered your suggestions and made some corrections which we hope meet with approval. Appended to this letter is our point-by-point response to the comments raised by the reviewers. The comments are reproduced and our responses are given directly afterward in a different color (red). Please find my itemized responses below and my revisions in the re-submitted files. We would like also to thank you for allowing us to resubmit a revised copy of the manuscript.

Point 1: The scientific names should be italicized throughout the manuscript.

Response 1: We are very sorry for our negligence of the incorrect format, and we have rewritten the names of yeasts species using italic in the full text.

Point 2: All figures are too small and not legible to read. High-resolution images with greater size should be provided.

Response 2: According to your suggestion, we have updated all figures in higher resolution and greater size.

Point 3: Line 52: Should read as ‘we attempted to review…’

Response 3: We are very sorry for this grammar mistake, and this sentence was rephrased as “we attempted to review…” (Line 52 in the re-submitted manuscript) according to the comment.

Point 4: Line 87: should read as ‘demonstrated significant activity…’. Remove the word ‘brilliant’

Response 4: Thank you for the suggestion. We have modified the sentence as “demonstrated significant activity…” (Line 87 in the re-submitted manuscript) according to your advice.

Point 5: Table 2: How was the ‘in-hospital mortality’ different from 7-day and 30-day mortality. Is there an overlap in these numbers? It is surprising that nearly 51% of the patients infected with C. albicans died. This number is too high compared to reported statistics on the Candidemia death rate. The authors should further explain this significant number of deaths. I am assuming that the death rate is high because of underlying causes which were aggravated by Candidemia. Please explain.

Response 5: Thank you for pointing this out. These three kinds of mortality reflect the death of patients with candidemia from different perspectives. In fact, there is an overlap in 7-day, 30-day mortality, and in-hospital mortality. The numbers of in-hospital mortality include all patients who died within 7-days and most patients died within 30-days. Thus, no approximately 51% of the patients infected with C. albicans died, and so do patients infected with other Candida species. We have added an explanation in the footnote of Table 2 according to your comment.

Point 6: Lines 176-179: delete these lines.

Response 6: We are very sorry for our negligence of the irrelevant text, and we have deleted the journal instructions in these lines.

Once again, we thank you for the time you put in reviewing our paper and look forward to meeting your expectations. Since your inputs have been precious, in the eventuality of a publication, we would like to acknowledge your contribution explicitly.

Reviewer 2 Report

Review of the article: Epidemiology, clinical characteristics, risk factors and outcomes of candidemia in a large tertiary teaching hospital in western China: a retrospective 5-year study from 2016 to 2020

Manuscript ID: antibiotics-1712183

In my opinion, the proposed manuscript is very interesting and well prepared. The performed analysis provide many information/conclusions that are important for both scientists and also medical doctors who work in hospitals or clinics. The most important advantages of the manuscript are:

  1. Interesting and really important subject of the study;
  2. Consideration of large number of isolates (episodes of infections);
  3. Using advanced methods of identification of Candia species and determination of MIC values (Sensititre YeastOneTM);
  4. Consideration many aspects that can be important for infection (baselinecharacteristics of patiens; risk factors; clinical laboratory data of patients);
  5. Fully professional statistical analysis;
  6. Carefully prepared manuscript;
  7. Obtained results are interesting from scientific and practical points of view.

I have also two critical remarks:

  1. The authors should go through the whole text and write names of yeasts species in agreement with requirements – using italic;
  2. In Table 1 susceptibility to AmB, 5-Flucytosine, Fluconazole, Itraconazole and Voriconazole was tested for 100 strains, whilst in the case of other antibiotics (namely Posaconazole, Caspofungin, Micaungin and Anidulafungin only 77 strains were considered. The same differences I found for other strains tested. I would be grateful for explanation.

In my opinion the manuscript can be accepted. Final decision - minor revision.  

Author Response

Dear Reviewer,

Thank you very much for your time involved in reviewing the previous version of the manuscript (Manuscript ID: antibiotics-1712183 and Title: Epidemiology, clinical characteristics, risk factors and outcomes of candidemia in a large tertiary teaching hospital in western China: a retrospective 5-year study from 2016 to 2020) and your very encouraging comments on the merits. We really appreciate all your comments and suggestions! These comments are all valuable and very helpful for revising and improving our paper, as well as the important guiding significance to our research. We have carefully considered your suggestions and made some corrections which we hope meet with approval. Appended to this letter is our point-by-point response to the comments raised by the reviewers. The comments are reproduced and our responses are given directly afterward in a different color (red). Please find my itemized responses below and my revisions in the re-submitted files. We would like also to thank you for allowing us to resubmit a revised copy of the manuscript.

Point 1: The authors should go through the whole text and write names of yeasts species in agreement with requirements – using italic;

Response 1: We are very sorry for our negligence of the incorrect format, and We have rewritten the names of yeasts species using italic in the full text.

Point 2: In Table 1 susceptibility to AmB, 5-Flucytosine, Fluconazole, Itraconazole and Voriconazole was tested for 100 strains, whilst in the case of other antibiotics (namely Posaconazole, Caspofungin, Micafungin and Anidulafungin only 77 strains were considered. The same differences I found for other strains tested. I would be grateful for explanation.

Response 2: Thank you for underlining this deficiency. These differences result from 2 antifungal susceptibility testing methods performed at different times. ATB FUNGUS 3 strips (bioMérieux, France) lacking Posaconazole, Caspofungin, Micafungin and Anidulafungin were performed from January 2016 to May 2017; Sensititre YeastOneTM antifungal panel (Thermo Scientific, USA) consisting of 9 antifungal agents were performed after April 2017; so, we lacked data on Posaconazole and echinocandins in a few months. We have described this limitation in the footnote of Table 1 and the “discussion” section according to the comment.

Once again, we thank you for the time you put into reviewing our paper and look forward to meeting your expectations. Since your inputs have been precious, in the eventuality of a publication, we would like to acknowledge your contribution explicitly.

Reviewer 3 Report

Overview

Hou, et al. present the epidemiology, species, and susceptibility testing related to over 250 candidemia cases at a teaching hospital in China. The paper focuses on species differences, especially between C. albicans and non-albicans Candida candidemia. This is a topic well-worth exploring given the recent rise in non-albicans Candida, leading to global concerns over antifungal resistance, and this investigation provides an abundance of details to examine this. However, the paper does need some essential editing for cleanliness and clarity (e.g., at one point the journal instructions for the manuscript are included in the text). I am also concerned that the paper’s major findings and conclusions about C. tropicalis resistance maybe the result of trailing growth or AFST reading rather than true resistance given the extreme results found and lack of discussion formally addressing that possibility. Please find the following specific questions and feedback:

General

  • Check for italicization, spelling, spacing, and capitalization consistency
  • Some of the figures are so small that they are practically unreadable.
  • Would recommend adding spaces to the tables as without spaces, it is challenging to read.

Introduction

  • Line 52: Incorrect phrasing: “…we attempt reviewed the microbial…”

Results

  • Line 77: Those less common NAC isolates add up to 11, or 4% of the overall count, not <2%
  • Line 78: 1/259 = 0.4%, as you had in the figure, but not the text for both C. lusitaniae and C. haemulonii.
  • Figure 2: A pie chart must add up to 100%, so a different method is needed to depict the small 4% NAC slice. If it still shows the 4% slice intact in the large pie, that color should have an associated label in the key.
  • Figure 2: Says C. lusitae instead of C. lusitaniae
  • Figure 3: Axis label says spices instead of species
  • Figure 3: I would recommend making it clearer that the p-value test is solely for overall C. albicans v. NAC. It is difficult to tell that now because the NAC species are visually grouped together but not labeled as being cohesively NAC and are instead broken down by each species.
  • Line 88-89: I would recommend clarifying this sentence as I had to read it multiple times to understand the intent. Perhaps something like the following: “More than 96% of Candida were susceptible to 3 echinocandins with the highest MIC50 of any species being ≤ 1 μg/mL and the highest MIC90 of any species being ≤ 2 μg/mL.”
  • Table 1: This is a suspiciously high azole resistance for C. tropicalis. It is well-known that C. tropicalis AFST readings can suffer from trailing growth. I wonder if much of the resistance here is due to trailing growth rather than true resistance. This should be critically examined before publication, especially because this is presented as one of the main findings for the abstract and discussion.
  • Table 1: It would be clearer for readers if non-existent categories were treated with n/a or “-“ instead of 0s (e.g. susceptible for C. glabrata).
  • Table 1: It’s hard to interpret the other category’s MICs because it’s a mix of different species. Perhaps it may be better to either write these out in text or in the footnote. At minimum, should clarify which isolates are included in each drug that had less than the full 11 other isolates tested.
  • Line 96-97: This sentence says 27 C. tropicalis isolates were resistant to voriconazole, but the table only shows 26.
  • Line 115-119: This sentence appears to be referring to table 4, not table 2, as these details are not presented in table 2. Is this in the wrong place within the text? Or do other details need to be added to table 2?
  • Line 123-125: This refers to the length of ICU stay, which is not presented in the table. Can this either be added to the table or to the text?
  • Table 2: Would not put decimals in the age results as all are whole numbers
  • Table 3: Just as was done in table 1, it seems table 3 needs a footnote and/or calculation adjustment given C. krusei is intrinsically resistant to fluconazole
  • Table 3: The footnote appears to be accidentally copied from table 2 instead of being the one for table 3 (i.e. it explains table 2 abbreviations but not those in table 3)
  • Figure 4: More explanation is needed for figure 4. For instance, is graph A counting all the NAC isolates or just those that are not C. tropicalis, C. parapsilosis, or C. glabrata?

Discussion

  • Lines 176-179: These appear to be the instructions for the discussion section from the journal
  • Line 274-275: Here and throughout, I would not consider this a retrospective study as all the information used in this study was recorded in the lab information system and medical charts in real time and therefore not subject to retrospective bias. It is simply that the charts were reviewed later.

Author Response

Dear Reviewer,

Thanks very much for taking your time to review the previous version of the manuscript (Manuscript ID: antibiotics-1712183 and Title: Epidemiology, clinical characteristics, risk factors and outcomes of candidemia in a large tertiary teaching hospital in western China: a retrospective 5-year study from 2016 to 2020). We are very grateful to the Reviewer for reviewing the paper so carefully. We really appreciate all your comments and suggestions! These comments are all valuable and very helpful for revising and improving our paper, as well as the important guiding significance to our research. We have carefully considered your suggestions and made some corrections which we hope meet with approval. Appended to this letter is our point-by-point response to the comments raised by the reviewers. The comments are reproduced and our responses are given directly afterward in a different color (red). Please find my itemized responses below and my revisions in the re-submitted files. We would like also to thank you for allowing us to resubmit a revised copy of the manuscript.

General

Point 1: Check for italicization, spelling, spacing, and capitalization consistency

Response 1: We are very sorry for our negligence of the incorrect format, and We have checked the consistency throughout the manuscript.

Point 2: Some of the figures are so small that they are practically unreadable.

Response 2: According to your suggestion, we have updated all figures in higher resolution and greater size.

Point 3: Would recommend adding spaces to the tables as without spaces, it is challenging to read.

Response 3: Thank you for the advice you gave me on tables modifications. We have added spaces to all tables.

Introduction

Point 4: Line 52: Incorrect phrasing: “…we attempt reviewed the microbial…”

Response 4: We are very sorry for this grammar mistake, and this sentence was rephrased as “we attempted to review…” (Line 52) according to the comment.

Results

Point 5: Line 77: Those less common NAC isolates add up to 11, or 4% of the overall count, not <2%

Response 5: Thank you for pointing this out. We have changed the sentence to be “The other uncommon NAC was almost less than 5% of all isolates comprising…” (Line 77).

Point 6: Line 78: 1/259 = 0.4%, as you had in the figure, but not the text for both C. lusitaniae and C. haemulonii.

Response 6: Thank you for pointing this out. We have updated the correct percent in the text, including C. lusitaniae (1/259, 0.4%) and C. haemulonii (1/259, 0.4%) (Line 78).

Point 7: Figure 2: A pie chart must add up to 100%, so a different method is needed to depict the small 4% NAC slice. If it still shows the 4% slice intact in the large pie, that color should have an associated label in the key.

Response 7: We would appreciate your advice on figures modifications. We have adopted a stacked bar chart to depict the small 4% NAC slice.

Point 8: Figure 2: Says C. lusitae instead of C. lusitaniae

Response 8: We are very sorry for our negligence of the incorrect spelling, and We have modified “C. lusitae” as “C. lusitaniae”.

Point 9: Figure 3: Axis label says spices instead of species

Response 9: We are very sorry for our negligence of the incorrect spelling, and We have modified “spices” as “species”.

Point 10: Figure 3: I would recommend making it clearer that the p-value test is solely for overall C. albicans v. NAC. It is difficult to tell that now because the NAC species are visually grouped together but not labeled as being cohesively NAC and are instead broken down by each species.

Response 10: We would appreciate your advice on figures modifications. We have specially marked NAC species in figure 3.

Point 11: Line 88-89: I would recommend clarifying this sentence as I had to read it multiple times to understand the intent. Perhaps something like the following: “More than 96% of Candida were susceptible to 3 echinocandins with the highest MIC50 of any species being ≤ 1 μg/mL and the highest MIC90 of any species being ≤ 2 μg/mL.”

Response 11: We are grateful for your advice on the sentence, and this sentence was rephrased as “More than 96% of Candida were susceptible to 3 echinocandins with the highest MIC50 of any species being ≤ 1 μg/mL and the highest MIC90 of any species being ≤ 2 μg/mL.” (Line 88-89) according to the comment.

Point 12: Table 1: This is a suspiciously high azole resistance for C. tropicalis. It is well-known that C. tropicalis AFST readings can suffer from trailing growth. I wonder if much of the resistance here is due to trailing growth rather than true resistance. This should be critically examined before publication, especially because this is presented as one of the main findings for the abstract and discussion.

Response 12: Thank you for pointing this out. Our feedbacks are as follows:

  1. Antifungal susceptibility testing was performed using ATB FUNGUS 3 strips (bioMérieux, France) and Sensititre YeastOneTM antifungal panel (Thermo Scientific, USA) in our study. Compared with ATB automated readings, ATB visual reading could significantly reduce the major errors in azole drug susceptibilities due to the trailing effect of azoles[1], and the strips were read visually all the time in our laboratory according to the practical algorithm[1]. YeastOneTM panel is a colorimetric microdilution test with an oxidation-reduction growth indicator, and the yeast growth will be evident as a color change, which alleviates some major concerns with the interpretation of certain Candida species because of 'trailing'[2].
  2. Our hospital has participated in a laboratory-based, nationwide multicenter surveillance study in China—CHIF-NET. Recently, the CHIF-NET surveillance program demonstrated that the continual decrease in the rate of azole susceptibility among C. tropicalis strains has become a nationwide challenge in China[3].
  3. Our clinical microbiology laboratory was accredited by the American CAP. The standard operating procedures of AFST were strictly implemented by well-trained laboratory personnel. Candida parapsilosis ATCC 22019 and Candida krusei ATCC 6258 were used as routine quality control for AFST, and all quality control results were within expected ranges. To avoid the potentially confounding effects of trailing growth on azole MIC, the MIC results of the same AFST test were independently determined by at least two different laboratory personnel after 24h incubation.

We have carefully reviewed all the antifungal susceptibility testing (AFST) data again, and it does indicate the high azole resistance rate of C. tropicalis.

Point 13: Table 1: It would be clearer for readers if non-existent categories were treated with n/a or “-“ instead of 0s (e.g. susceptible for C. glabrata).

Response 13: We would appreciate your advice on tables modifications. Following your suggestions, we have modified the table.

Point 14: Table 1: It’s hard to interpret the other category’s MICs because it’s a mix of different species. Perhaps it may be better to either write these out in text or in the footnote. At minimum, should clarify which isolates are included in each drug that had less than the full 11 other isolates tested.

Response 14: We would appreciate your advice on tables modifications and agree that it’s hard to interpret the other category’s MICs. We have added more details in the footnote of Table 1 following your suggestions.

Point 15: Line 96-97: This sentence says 27 C. tropicalis isolates were resistant to voriconazole, but the table only shows 26.

Response 15: We are very sorry for our negligence of the inconsistent numbers, and the statements “and most voriconazole-resistant C. tropicalis were resistant (26/27, 96.3%) to fluconazole” were corrected as “and most voriconazole-resistant C. tropicalis were resistant (25/26, 96.2%) to fluconazole” (Line 97-98) after a careful examination.

Point 16: Line 115-119: This sentence appears to be referring to table 4, not table 2, as these details are not presented in table 2. Is this in the wrong place within the text? Or do other details need to be added to table 2?

Response 16: Thank you for pointing this out. We have moved the text “Of note, compared with C. albicans candidemia, the prevalence of hematological dis-order was markedly more frequent in patients with C. tropicalis candidemia, while the proportions of invasive procedures with significant differences, such as surgery, invasive mechanical ventilation, urinary catheter, and indwelling thoraco-abdominal drainage catheters, were slightly lower in the NAC group.” to the “2.4. C. albicans vs. non-albicans Candida” section (Line 150-154).

Point 17: Line 123-125: This refers to the length of ICU stay, which is not presented in the table. Can this either be added to the table or to the text?

Response 17: We are very sorry for our negligence of the lack of data, and we have added the relevant data in table 2 and have redescribed the statement (Line 127-129).

Point 18: Table 2: Would not put decimals in the age results as all are whole numbers

Response 18: We are grateful for your advice on tables modifications. We have removed the decimals in the age results.

Point 19: Table 3: Just as was done in table 1, it seems table 3 needs a footnote and/or calculation adjustment given C. krusei is intrinsically resistant to fluconazole

Response 19: Thank you for pointing this out. We have added “C. krusei were inherently resistant to fluconazole” in the footnote of Table 3.

Point 20: Table 3: The footnote appears to be accidentally copied from table 2 instead of being the one for table 3 (i.e. it explains table 2 abbreviations but not those in table 3)

Response 20: Thank you for pointing this out. We are very sorry for our negligence of content matching, and we have corrected the footnote of Table 3.

Point 21: Figure 4: More explanation is needed for figure 4. For instance, is graph A counting all the NAC isolates or just those that are not C. tropicalis, C. parapsilosis, or C. glabrata?

Response 21: We are grateful for your advice on figures modifications. We have updated figure 4 and have added more explanation in the footnote according to your comment.

Discussion

Point 22: Lines 176-179: These appear to be the instructions for the discussion section from the journal

Response 22: Thank you for pointing this out. We are very sorry for our negligence of the irrelevant text, and we have deleted the journal instructions in these lines.

Point 23: Line 274-275: Here and throughout, I would not consider this a retrospective study as all the information used in this study was recorded in the lab information system and medical charts in real time and therefore not subject to retrospective bias. It is simply that the charts were reviewed later.

Response 23: Thank you for pointing this out. Although we reviewed all information of patients from the laboratory and hospital information system, we think that the present study would meet the definition of a retrospective study. A retrospective study is performed a posteriori, using the information on events that have taken place in the past. By the way, the baseline state, diagnosis, and intervention are obtained from pre-existing information that was recorded for reasons other than the study. Our study remains limited by the accuracy, completeness and comprehensiveness of data in the patient electronic medical records.

Once again, we thank you for the time you put into reviewing our paper and look forward to meeting your expectations. Since your inputs have been precious, in the eventuality of a publication, we would like to acknowledge your contribution explicitly.

Reference

  1. Zhang, L.; Wang, H.; Xiao, M.; Kudinha, T.; Mao, L.L.; Zhao, H.R.; Kong, F.; Xu, Y.C. The widely used ATB FUNGUS 3 automated readings in China and its misleading high MICs of Candida spp. to azoles: challenges for developing countries' clinical microbiology labs. PLoS One 2014, 9, e114004, doi:10.1371/journal.pone.0114004.
  2. SENSITITRE YEASTONE For in vitro Diagnostic Use. Available online: https://docplayer.net/21149741-Sensititre-yeastone-for-in-vitro-diagnostic-use.html (accessed on 18 May 2022).
  3. Wang, Y.; Fan, X.; Wang, H.; Kudinha, T.; Mei, Y.N.; Ni, F.; Pan, Y.H.; Gao, L.M.; Xu, H.; Kong, H.S.; et al. Continual Decline in Azole Susceptibility Rates in Candida tropicalis Over a 9-Year Period in China. Front Microbiol 2021, 12, 702839, doi:10.3389/fmicb.2021.702839.